# The Influence of *FAM13A* and *PPAR-γ2* Gene Polymorphisms on the Metabolic State of Postmenopausal Women

**DOI:** 10.3390/genes14040914

**Published:** 2023-04-14

**Authors:** Bogna Grygiel-Górniak, Iwona Ziółkowska-Suchanek, Lidia Szymkowiak, Natalia Rozwadowska, Elżbieta Kaczmarek

**Affiliations:** 1Department of Rheumatology, Rehabilitation and Internal Diseases, Poznan University of Medical Sciences, 61-701 Poznan, Poland; 2Institute of Human Genetics, Polish Academy of Sciences, 60-479 Poznan, Poland; 3Department of Bioinformatics and Computational Biology, Poznan University of Medical Sciences, 60-806 Poznan, Poland

**Keywords:** obesity, immunogenetics, hypertension, *PPAR-γ2*, *3β-AR*, *FAM13A*, polymorphisms

## Abstract

Recently, we have observed two significant pandemics caused by communicable (COVID-19) and non-communicable factors (obesity). Obesity is related to a specific genetic background and characterized by immunogenetic features, such as low-grade systemic inflammation. The specific genetic variants include the presence of polymorphism of the Peroxisome Proliferator-Activated Receptors gene (*PPAR-γ2*; Pro12Ala, rs1801282, and C1431T, rs3856806 polymorphisms), β-adrenergic receptor gene (*3β-AR*; Trp64Arg, rs4994), and Family With Sequence Similarity 13 Member A gene (*FAM13A*; rs1903003, rs7671167, rs2869967). This study aimed to analyze the genetic background, body fat distribution, and hypertension risk in obese metabolically healthy postmenopausal women (*n* = 229, including 105 lean and 124 obese subjects). Each patient underwent anthropometric and genetic evaluations. The study has shown that the highest value of BMI was associated with visceral fat distribution. The analysis of particular genotypes has revealed no differences between lean and obese women except for *FAM13A* rs1903003 (CC), which was more prevalent in lean patients. The co-existence of the *PPAR-γ2* C1431C variant with other *FAM13A* gene polymorphisms [rs1903003(TT) or rs7671167(TT), or rs2869967(CC)] was related to higher BMI values and visceral fat distribution (WHR > 0.85). The co-association of *FAM13A* rs1903003 (CC) and *3β-AR* Trp64Arg was associated with higher values of systolic (SBP) and diastolic blood pressure (DBP). We conclude that the co-existence of *FAM13A* variants with C1413C polymorphism of the *PPAR-γ2* gene is responsible for body fat amount and distribution.

## 1. Introduction

The prevalence of obesity worldwide is increasing every year, and over 1.9 billion adults were overweight or obese in 2016 [1,2]. Recent projections show that nearly 50% of adults in the USA will be obese by 2030, and there are similar predictions regarding global obesity [3]. The most important and prevalent consequence of obesity is hypertension, a major cardiovascular disease risk factor. Elevated blood pressure or already diagnosed hypertension accounted for 9.3% of disability-adjusted life-years lost globally and 10.8 million deaths (19.2% of all deaths in 2019) [4]. Spanish data show that BMI (Body Mass Index) increases with age, and obesity is more common in postmenopausal women than men (78.4% vs. 48.4%, respectively) [5].

In obesity, low-grade inflammation is observed. Both adaptive and innate immune cell-mediated inflammation can occur in metabolically active fat tissue, particularly visceral adipocytes [6,7]. In excessive fat accumulation, macrophages and other immune cells infiltrate adipose tissues, shifting their anti-inflammatory profile to pro-inflammatory properties [8]. In healthy older people, chronic inflammatory dysregulation is associated with increased inflammatory markers (e.g., C-reactive protein, interleukin-6, TNF-α, interleukin-1β) typical for various cardiovascular diseases and diabetes mellitus [8,9]. Excessive pro-inflammatory cytokines adversely influence vascular function throughout the body [10]. Thus, obesity is considered an excessive adipose tissue deposition, which is strictly related to immune dysregulation, endothelial dysfunction, and metabolic consequences. Moreover, the secretion of various inflammatory mediators stimulates the development of cardiovascular diseases (including hypertension), making them more resistant to treatment [2].

Recent studies have described the immunogenetic background of obesity, which increases after menopause and in the elderly [11,12,13,14]. They also showed the relevance of immunogenetic markers (HLA—human leucocyte antigens, cytokine genes, and some innate immunity genes) for successful aging and an increased life span. Thus, the immunogenetic aspects of obesity seem to play a crucial role in elderly society [15].

One of the obesity-related genes is the *PPAR-γ2* gene, a transcription factor that modulates the expression of several genes responsible for fatty acid metabolism, glucose homeostasis, and insulin sensitivity [16,17]. *PPAR-γ2* polymorphisms (Pro12Ala, rs1801282 and C1431T, rs3856806) and β-adrenergic receptor gene polymorphisms (*3β-AR* Trp64Arg, rs4994) influence the adipose tissue activity, metabolic pathways and endothelial inflammation, and determine high blood pressure [18,19,20,21].

Other genetic variants associated with obesity are *FAM13A* gene (Family With Sequence Similarity 13 Member A) polymorphisms. They are related to normal or lower adiposity but have a poorer cardiometabolic profile resembling a metabolically unhealthy normal weight phenotype [22]. The study of M. Fathzadeh et al. revealed that in *FAM13A* gene variants, a lower BMI is associated with a higher waist-to-hip ratio (WHR) and visceral-to-subcutaneous adipose tissue ratio (VAT/SAT) [22]. Unfortunately, the biological function of the *FAM13A* gene product is poorly understood. The highest expression of *FAM13A* is detected in the adipose tissue, brain, ovaries, and pancreas, followed by the lungs and kidneys [23,24]. This gene encodes a protein with two coiled-coil domains and three nuclear localization signals. The most important part of the FAM13A protein is its N-terminal extension, containing the Rho-GAP domain. Previously, *FAM13A* gene variants were indicated as risk factors for chronic lung diseases, including chronic obstructive pulmonary disease (COPD) [25,26], lung cancer [26,27,28,29], and cystic fibrosis [30]. The comprehensive study of M. Fathzadeh et al. presented the biological mechanism linking FAM13A with body fat distribution based on genetics studies and in vivo and in vitro experiments. GWAS-associated (Genome-Wide Association Study) variants within the *FAM13A* locus alter adipose FAM13A expression, which regulates adipocyte differentiation and contributes to changes in body fat distribution [22]. The study by Lin et al. revealed that the rs2276936 locus of the *FAM13A* gene has allelic-enhancer activity and regulates the expression of endogenous FAM13A in hepatocellular carcinoma (HepG2) cells. Moreover, Fam13a−/− mice were protected from high-fat diet-induced fatty liver accompanied by increased insulin sensitivity and reduced glucose production in the liver [31].

Nevertheless, the biological function of FAM13A and the molecular mechanism of its action on the distribution of fat in healthy subjects and cardiometabolic diseases has not been fully elucidated. The influence of *FAM13A* gene variants on the metabolic state of non-smoking metabolically healthy postmenopausal women had not been studied before. In the current study, we aimed to assess the association of selected *FAM13A* SNPs with other genes, such as *PPAR-γ2* and *3β-AR*, with body fat distribution and the risk of hypertension in obese postmenopausal women.

## 2. Materials and Methods

### 2.1. Selection Criteria and Ethical Aspects

A total of 877 postmenopausal women were selected from the Metabolic Outpatient Clinic and invited to undergo anthropometrical measurements as described in our previous paper [32]. This study group included women without essential diseases such as non-treated thyroid disorders, acute liver and renal diseases, neoplasms diagnosed during the last five years, cardiovascular diseases (except for patients with arterial hypertension), acute infections, smoking, and did not include women taking vitamins or mineral supplements. Subjects receiving concomitant medication, including lipid-lowering and hypoglycemic medications, were excluded from this analysis. Finally, an examination and interview with a physician enabled 229 women to be selected for the study (105 lean women; BMI < 25 kg/m^2^) and obese (*n* = 124; BMI > 30 kg/m^2^). All subjects enrolled in this study provided their written informed consent. This study was approved by the local Bioethics Committee of Poznan University of Medical Sciences (no. 792/09) and was performed according to the Helsinki Declaration.

### 2.2. Data Collection

All volunteers were evaluated at the same time of day to avoid different responses of physiological variables due to circadian changes. Basic anthropometric measurements, bioimpedance, and blood pressure measurements were collected during the fasting stage. All anthropometric and blood pressure parameters were measured twice by study staff, using a standardized protocol, and averaged.

### 2.3. Blood Pressure Measurement

Blood pressure was measured accurately to classify individuals as normo- or hypertensive and to ascertain blood pressure-related risks in the office. Blood pressure was measured using a mercury sphygmomanometer, which is widely regarded as the “gold standard” for office blood pressure measurement [33]. The blood pressure was taken between 7:00 am and 11:00 am after an overnight fast, by a physician using a stethoscope and listening for the Korotkoff sounds [34]. The measurement was carried out on the patient’s non-dominant arm after a 10 min rest in the upright seated position. Hypertension was defined as systolic blood pressure (SBP) ≥ 140 mmHg and/or diastolic blood pressure (DBP) ≥ 90 mmHg according to the recent guidelines of the European Society of Cardiology and the European Society of Hypertension [35].

### 2.4. Basic Anthropometric Measurements

A calibrated digital platform scale measured the current body mass with a capacity of 200 kg, and 100 g accuracy was used (bare feet, in light clothing). Height was determined with a vertical ruler with an accuracy of 0.1 cm. Waist circumference was measured to the nearest 0.1 cm, midway between the lower border of the ribs and the iliac crest at the widest portion. Hip circumference was obtained to the nearest 0.1 cm at the widest diameter of the buttocks. Body mass and height data were used to calculate BMI (body weight/body height squared; kg/m^2^) and WHR (waist-to-hip ratio), later classified according to WHO (2000) [36]. A high WHR > 0.85 in women indicated visceral (abdominal) fat accumulation. Anthropometric measurements were made according to the current recommendations [37].

### 2.5. Bioimpedance

The bioimpedance method assessed the body fat and lean body mass (LBM) using BODYSTAT 1500—a single-frequency device (50 kHz; Bodystat Ltd., Douglas, Isle of Man, UK).

### 2.6. Genetic Methods

#### 2.6.1. DNA Isolation

According to the manufacturer’s instructions, DNA samples from selected patients were isolated from peripheral blood lymphocytes by a Gentra Puregene Blood Kit (Qiagen, Hilden, Germany). DNA purity and concentration were confirmed using a NanoDrop ND-1000 spectrophotometer. All DNA samples were diluted to 5 ng/μL for genotyping reactions.

#### 2.6.2. SNP Genotyping

Data associating *PPAR-γ2* (Pro12Ala, rs1801282, and C1431T, rs3856806) and *3β-AR* (Trp64Arg, rs4994) polymorphic sites with metabolic diseases were obtained from our previous studies [32]. For the *FAM13A* gene, we selected the SNPs based on a literature review and used the most significant reported polymorphisms, which were analyzed in relatively large groups of patients and had functional potential. The chosen SNPs (rs1903003, rs7671167, rs2869967) were previously identified by M. Cho [25]. To achieve sufficient statistical power, all polymorphisms selected for this study had minor allele frequencies > 0.4. The SNPs were genotyped using TaqMan^®^ SNP genotyping assays (Life Technologies, Carlsbad, California; assays IDs: *FAM13A*: 1143659_10, 1143656_10, 15837681_10). The PCR was performed with HOT FIREPol Probe qPCR Mix Plus (no ROX) according to the manufacturer’s instructions provided by Solis Biodyne (Tartu, Estonia). The PCR thermal conditions were as follows: initial denaturation at 95 °C for 15 min; 40 cycles of 95 °C for 15 s and 60 °C for 60 s. Thermal cycling was performed using a CFX96 Touch™ Real-Time PCR Detection System (Bio-Rad, Hercules, CA, US). Negative controls and approximately 5% of the samples were genotyped in duplicate as a quality control measure to check the genotyping accuracy. The genotypes of selected samples were confirmed by direct sequencing (Genomed S. A., Warszawa, Poland).

### 2.7. Statistical Analysis

The distribution of selected polymorphisms PPAR-γ2 (Pro12Ala, rs1801282 and C1431T, rs3856806), β-adrenergic receptor gene (3β-AR; Trp64Arg, rs4994), and Family With Sequence Similarity 13 Member A gene (FAM13A; rs1903003, rs7671167, rs2869967) was analyzed using Pearson’s chi-square test or Fisher’s exact test (for small frequencies). The expected frequencies were determined by the Hardy–Weinberg equilibrium (p^2^ + 2pq + q^2^ = 1, where p is the frequency of the variant allele and q = 1 − p). Frequencies of analyzed polymorphisms between lean and obese women were verified by chi-^2^ test, and odds (ORs) ratios with 95% confidence intervals (95%CI) were determined.

The results of the study (continuous variables) were subjects of descriptive statistics (mean +/− SD) in two independent groups: normal body mass (BMI < 25, *n* = 105 women) and obesity (BMI > 30, *n* = 124). Multiphenotype association testing was performed to find significant polymorphisms/genes variants for the studied groups. Univariate test statistics of SNPs of multiple genetic variants were combined to compute statistics based on a multivariate normal distribution with the mean (combined z scores of all SNPs) and a covariance matrix of the genetic variants. The covariance matrix was approximated by the sample covariance matrix of the Z scores of all SNPs. The Holm–Bonferroni correction was used in the analysis of multiple associations. This part of the analysis was implemented in R-Studio. Significant genetic variants with frequencies of 5 or more were selected for further analysis.

The Mann–Whitney U test was also used to compare the distributions of quantitative continuous variables in two studied groups in detected significant genetic variants.

A difference between compared results was accepted as significant at *p* < 0.05. The statistical analysis was performed by using Statistica software (StatSoft, Inc., Tulsa, OK, USA) and an application originally designed in R-Studio.

## 3. Results

### 3.1. Characteristics of the Analyzed Group

The analyzed women were characterized by similar age and height and varied between groups with anthropometrical parameters according to the study design (Table 1). Obese women have higher WHR and blood pressure values (both systolic and diastolic).

### 3.2. Genotypes Prevalence

When the Hardy–Weinberg equilibrium was evaluated, we observed that all the polymorphism genotype distributions were in accordance with Hardy–Weinberg expectations in both groups (obese subjects and lean women). Thus, the study groups were representative (Table 2). The analysis of polymorphism frequency between lean and obese women showed that the *PAPR-γ2* Pro12Pro and C1431T, *β3-AR*: Trp12Arg, *FAM13A* rs7671167 and rs2869967 did not differ between groups. The differences between obese and lean women were observed only in the case of the *FAM13A* rs1903003 (CC) genotype.

### 3.3. Genotypes’ Co-Existence

The differences in the analyzed anthropometric values (higher BMI and WHR) were observed only in the case of co-existing *FAM13A* rs1903003 (TT) and *PPARγ-2* C1431C (Table 3). Similarly, the presence of *FAM13A* rs7671167 (TT) and C1431C, as well as the co-existence of *FAM13A* rs2869967 (CC) with C1431C, was associated with higher values of BMI and WHR. On the other hand, the occurrence of the rs1903003 (CC) genotype with C1431T was related to higher values of systolic arterial pressure. Higher values of SBP systolic blood pressure (SBP) and diastolic blood pressure (DBP) were observed in the case of *FAM13A* rs1903003 (CC) and co-existed with *β3-AR* Trp64Arg (such a relationship was not observed in the presence of the other analyzed parameters). We did not notice the differences between anthropometric parameters and blood pressure values in any of the *FAM13A* rs1903003 or rs7671167, or rs2869967 with the *PPARγ-2* Pro12Ala genotype. The co-existence of *β3-AR* gene variants with *FAM13A* gene polymorphisms (rs1903003, rs7671167, and rs2869967) showed no significant differences between C1431 and T1431 alleles on WHR values within the lean and obese groups except for genotype rs2869967 (CC), which was more frequent among women with lean body mass compared to the obese group (Table 4).

## 4. Discussion

The prevalence of obesity increases in the elderly population (including postmenopausal women). It is linked to systemic inflammation induced by adipokines and the activation of gene expression related to inflammation in adipose tissue [9]. Obesity often develops after menopause, which is related to immunogenetic background. Menopause (particularly premature) is associated with autoimmune or genetic diseases, infections, enzyme deficiencies, and metabolic syndrome [11]. Excessive fat tissue accumulation is associated with excessive pro-inflammatory cytokine synthesis, and causes regional inflammation [9]. Consequently, systemic low-grade chronic inflammation develops, increasing the risk of various autoimmune and metabolic diseases such as diabetes mellitus and cancer [38].

In obesity, enlarged adipocytes press the surrounding vessels and adipose tissue cells and cause local hypoxia [39]. As a result, increased fibrotic processes and the infiltration of macrophages are observed. Macrophages scavenge necrotic adipocytes and trigger a local inflammatory response. They increase the synthesis of pro-inflammatory cytokines such as TNF-a and IL-6 [39]. The inflammatory state in obesity is also related to the transformation of M2 macrophages (anti-inflammatory) to M1 macrophages (pro-inflammatory). Such a conversion correlates with insulin resistance [40]. Low-grade inflammation in adipose tissue is also stimulated by B-cells, which infiltrate the adipose tissue and synthesize various cytokines and antibodies (IgG) [40,41].

The highest pro-inflammatory activity of adipocytes is observed mainly in visceral fat distribution [42,43]. Conversely, subcutaneous adipose tissue is an energy-storage depot preventing fat accumulation in organs. In this study, obese women had higher waist-to-hip ratio values, indicating visceral fat distribution (Table 1). According to the International Diabetes Federation, central adiposity (waist circumference over 88 cm) is the component of metabolic syndrome [44]. In this study, waist circumference in obese women crossed 100 cm, and was much higher than in the lean group (Table 1). Excessive abdominal adipose tissue accumulation increased the risk of cardiometabolic disorders, including hypertension [45,46]. Indeed, in this study, SBP and DBP values were higher in postmenopausal obese women. Recent data confirm that elevated blood pressure, particularly in obesity, is a risk factor for cardiovascular events, which increases with age [1,2,47,48,49]. Blood pressure values of SBP ≥ 130 mmHg or DBP ≥ 85 mmHg with the increased WHR in obese subjects contribute to metabolic syndrome [44,50]. Thus, hypotensive therapy with caloric restriction should be implemented in these patients to reduce cardiometabolic risk. The prevalence of particular genotypes is shown in Table 2. In the whole group of analyzed Caucasian Polish women (both lean and obese), the following frequency of selected polymorphisms was observed: *PPAR-γ2:* Ala12 allele (0.147), T1431 allele (0.174); *β3-AR:* Arg64 allele (0.99); *FAM13A;* rs7671167 allele C (0.457), rs1903003 allele (0.518), and rs2869967 allele C (0.480). In recently published data, a similar allele distribution was observed concerning *PPAR-γ2:* Ala (0.11) [51], T1431 (0.160) [52]: *β3-AR*: Arg (0.123) [53]; *FAM13A:* rs7671167 allele C (0.51), rs2869967 allele C (0.42), and rs1903003 allele C (0.48) [28]. Until now, polymorphisms such as Pro12Ala, C1431T (both in *PPAR-γ2*), and Trp64Arg (*β3-AR*) were mainly associated with metabolic disorders, while *FAM13A* gene variants (rs1903003, rs7671167, rs2869967) were more prevalent in COPD [25,54,55]. However, in the Polish population, only rs2869967 among three *FAM13A* variants showed an association with an increased risk of COPD, whereas rs7671167 and rs1903003 were significantly more frequent among the controls [28]. Interestingly, some studies show that specific *FAM13A* loci may be associated not only with pulmonary disorders, but also with metabolic complications. Clinical studies have revealed that the *FAM13A* locus is involved in the genetic control of systemic insulin sensitivity [56], lipid levels (HDL, high-density lipoprotein) [57], and the value of the waist-to-hip ratio [58]. Thus, we intended to check whether these genotypes could be related to postmenopausal obesity, and investigate their association with arterial hypertension.

The analysis of selective polymorphic sites showed no differences in genotypes between lean and obese women, with the exception of *FAM13A* rs1903003 (CC) (Table 2). The prevalence of CC genotypes of rs1903003 was significantly elevated among the lean group compared with obese cases, suggesting this allele’s protective effect. However, the higher prevalence of this genotype can be related to the risk of hypertension, mainly if this genotype co-exists with other metabolic polymorphisms (*PPARγ2* C1431 variant, discussed below). Other studies confirm that the *FAM13A* locus alters adipose FAM13A expression, which regulates adipocyte differentiation and contributes to changes in body fat distribution [22]. Thus, the higher prevalence of the *FAM13A* rs1903003 C allele can modulate the amount and distribution of fat tissue.

Whether any of the selected *FAM13A* SNPs are functional remains unknown, although given their location, an effect on splice variants would be possible. All three variants (rs2869967, rs7671167, rs1903003) were described as missense, intronic, and beyond the potential enhancer region. The gene predictions and transcript consequences were nonsense-mediated mRNA decay (NMD) or non-coding transcript variant (data from https://www.ensembl.org/Homo_sapiens/Variation/, accessed on 23 January 2023).

Previous studies of SNPs rs7671167 and rs1903003 locate them in intron 4 of the *FAM13A* gene downstream of the most important part of the FAM13A molecule: the Rho-GAP domain. Although little is known about FAM13A function, the Rho GTPase activating role suggests both anti-inflammatory and tumor suppressor activity. Furthermore, other data suggest that Rho GTPase signaling regulates adipogenesis [22]. As with the other non-coding variants of *FAM13A*, e.g., rs3822072 and rs9991328, an association with higher fasting insulin levels adjusted for BMI and increased WHR adjusted for BMI was found. The biological function of these variants in this context is not fully understood and needs further evaluation.

Since the prevalence of fat content is not determined by a single SNP, we decided to check whether the co-existence of the analyzed *FAM13A* gene polymorphisms (rs1903003, rs7671167, and rs2869967) with *PPAR-γ2* gene variants (Pro12Ala, C1431T) and *β3-AR* (Trp64Arg) could determine the distribution of body fat in analyzed postmenopausal women (independently of BMI value) (Table 3). The *PPAR-γ2* and *β3-AR* genes’ polymorphisms are commonly described in various obesity-associated cardiometabolic disorders [59,60,61,62,63]. Additionally, previous studies have suggested that the co-occurrence of selective haplotypes (e.g., *PPAR-γ2* Pro12-T1341) is associated with increased body mass rather than particular SNP [52].

One of the gene variants which can influence the described anthropometric parameters is a common *PPAR-γ2* C1431 polymorphism, a synonymous CT substitution at nucleotide position 161 in exon 6 (C161T; CAC478CAT, rs3856806) [64]. In this study, the analysis of selected genotypes showed that the co-existence of *PPAR-γ2* C1431C polymorphism with *FAM13A* rs1903003 (TT) or rs7671167 (TT), or rs2869967 (CC), was related to higher values of BMI and WHR. Interestingly, variant C1431T of *PPAR-γ2* has been associated with obesity [52], hypertension, and preeclampsia risk [65]. Conversely, the *PPAR-γ2* T1431 variant has opposing associations with body mass [66]. The study of Liu et al. showed that T allele carriers of the *PPAR-γ2* rs3856806 polymorphism had a reduced coronary artery disease (CAD) risk compared with CC homozygotes [67]. Similar data were presented by Zhou et al., who showed that the T allele carriers of the *PPAR-γ2* rs3856806 polymorphism had a 39% decreased risk of CAD relative to CC homozygotes (Chinese population) [68]. Similarly, in this study, we proved that the presence of the *PPAR-γ2* C1431C variant with other *FAM13A* gene polymorphisms [rs1903003(TT) or rs7671167(TT), or rs2869967(CC)] was related to higher BMI values and visceral fat distribution (WHR > 0.85), and such associations can increase the cardiovascular risk.

Since the presence of C1431C polymorphism in the *PPAR-γ2* gene was associated with higher BMI and WHR values, we decided to check whether CT1431T or C1431C affected body fat distribution in the case of co-existence with *FAM13A* gene polymorphic variants in the group of lean and obese women. The data in Table 4 analyzing the specific *FAM13A* and *PPAR-γ2* polymorphisms showed no significant differences between *PPAR-γ2:* C1431 and T1431 alleles within lean and obese groups except for the *FAM13A* genotype rs2869967 (CC) in obese women. Thus, the influence of C1431C depends on body mass and is related to higher BMI values.

This study also showed that the co-association of *FAM13A* rs1903003 (CC) and the *β3-AR* Trp64 variant is related to higher blood pressure values (SBP and DBP). Thus, these data correspond with other studies which have proved that the Trp64Arg polymorphism of the *β3-AR* gene is associated with increased metabolic disorders (diabetes mellitus) and hypertension risk [69,70].

In summary, the presence of *FAM13A* and *PPAR-γ2* polymorphisms is related to the amount and distribution of fat tissue, which can determine the cardiovascular risk of analyzed women. Recently, Huang et al. identified 62 loci that support the key role of fat distribution for the *FAM13A* and *PPAR-γ2* genes, which confirmed the strong biological candidacy of these genes to link obesity to cardiometabolic health [71].

The co-existence of *FAM13A* and *PPAR-γ2* polymorphisms (particularly the C1431 allele) affects the amount and distribution of adipose tissue and blood pressure in postmenopausal obesity. The specific crosstalk between genotypes analyzed in this study may predispose an individual to quicker fat tissue accumulation, higher blood pressure values, and a worse response to treatment (diet therapy and pharmacological treatment). Therefore, such patients should remain under special medical and dietary supervision to carefully monitor the effects of weight loss and blood pressure control.

## 5. Conclusions

This study’s analysis of specific polymorphisms has shown that increasing body weight is related to visceral adipose tissue distribution and specific genotypes of *FAM13A* and *PPAR-γ2* genes. The prevalence of *FAM13A* rs1903003 (CC) was higher in postmenopausal lean women. The analyzed data indicated that the co-occurrence of *FAM13A* polymorphisms (rs1903003, rs7671167, and rs2869967) with the *PPAR-γ2* C1431C variant in postmenopausal women was associated with higher values of anthropometric parameters (BMI and WHR). Thus, such an allelic association (*FAM13A* gene variants with C1431T *PPAR-γ2* polymorphism) could influence the amount and distribution of body fat after menopause. In contrast, the presence of *FAM13A* rs1903003 (CC) with Trp64Arg of the *β3-AR* gene was related to higher values of SBP and DBP. However, it is also possible that these polymorphisms differentially mark separate haplotypes associated with a distinct phenotypic effect, which has not been explored to date. For the better prevention and treatment of obesity, more attention should be paid to anthropometric indices and genetic predisposition. Postmenopausal women should be treated actively, particularly those with *FAM13A* rs1903003 (CC) co-existing with *PPAR-γ2* C1431T polymorphism. The specific crosstalk amongst various genotypes under obese conditions can determine the effectivity of body mass reduction and hypertension treatment. *FAM13A* SNPs variants should be considered as a potential diagnostic tool in the case of postmenopausal women. However, further studies are required to elucidate the functional role of these variants, which may have an impact on fat tissue disruption and the development of metabolic disorders. The directions should be focused on a broad approach to finding metabolic complications in obese postmenopausal women, and implementing the most effective treatment to keep a good health status.

## Figures and Tables

**Table 1 genes-14-00914-t001:** Characteristics of analyzed groups of lean and obese postmenopausal women.

Characteristics of Analyzed Groups			
Analyzed Parameters	Normal Body Mass *n* = 105	Obesity *n* = 124	
	X ± SD	X ± SD	*p*-Value
Age [years]	58.91 ± 5.65	59.57 ± 5.01	0.3444
Height [cm]	161.80 ± 5.84	160.26 ± 6.01	0.0507
Body mass [kg]	62.10 ± 6.91	87.27 ± 11.91	0.00001
Waist circumference [cm]	77.67 ± 7.85	100.15 ± 9.21	0.00001
Hip circumference [cm]	97.62 ± 5.40	115.19 ± 8.45	0.00001
WHR	0.80 ± 0.07	0.87 ± 0.06	0.00001
FBM [% body mass]	37.62 ± 4.02	47.73 ± 4.14	0.00001
LBM [% body mass]	62.44 ± 3.89	52.23 ± 4.14	0.00001
BMR [kcal/day]	1277.34 ± 103.10	1431.23 ± 114.33	0.00001
SBP [mmHg]	137.06 ± 21.35	145.27 ± 23.68	0.0068
DBP [mmHg]	84.38 ± 12.71	91.27 ± 13.57	0.0001

WHR—waist-to-hip circumference, LBM—lean body mass [% of total body mass], FBM—fat body mass [% of total body mass], BMR—basal metabolic rate, SBP—systolic blood pressure, DBP—diastolic blood pressure.

**Table 2 genes-14-00914-t002:** Frequency of analyzed *PAPR-γ2*, *β3-AR*, and *FAM13A* polymorphisms between lean and obese women.

Gene SNPs	Name	Frequency (%)		OR (95% CI)	*p* Value
		Lean *n* = 105	Obese *n* = 124		
** *PAPR-γ2* **					
**rs1801282**	**Pro12Ala**				
Genotypes					
CC	Pro12Pro	74 (70.5)	93 (75)	1 ^1^	
CG	Pro12Ala	30 (28.5)	27 (21.8)	0.7161 [0.3918–1.309]	0.2768
GG	Ala12Ala	1 (1)	4 (3.2)	3.183 [0.348–29.11]	0.2801
Alleles					
C	Pro	178 (84.7)	213 (85.9)	1 ^1^	
G	Ala	32 (15.3)	35 (14.1)	0.9140 [0.5439–1.536]	0.7342
**rs3856806**	**C1431T**				
Genotypes					
CC	C1431C	71 (67.6)	84 (67.7)	1 ^1^	
CT	C1431T	33 (31.4)	35 (28.2)	0.8965 [0.5064–1.587]	0.7075
TT	T1431T	1(1)	5 (4.1)	4.226 [0.4822–37.0]	0.1590
Alleles					
C	C1431	175 (83.3)	203 (81.9)	1 ^1^	
T	T1431	35 (16.7)	45 (18.1)	1.108 [0.6818–1.802]	0.6780
* **β3-AR** *					
**rs4994**	**Trp12Arg**				
Genotypes					
TT	Trp12Trp	84 (80)	102 (82.2)	1 ^1^	
TC	Trp12Arg	20 (19)	21 (17)	0.8647 [0.4393–1.702]	0.6737
CC	Arg12Arg	1 (1)	1 (0.8)	0.8235 [0.05071–13.3]	0.8912
Alleles					
T	Trp	188 (89.5)	225 (90.7)	1 ^1^	
G	Arg	22 (10.5)	23 (9.3)	0.8735 [0.4718–1.617]	0.6667
* **FAM13A** *					
**rs7671167**					
Genotypes					
TT		33 (31.4)	32 (25.8)	1 ^1^	
TC		52 (49.5)	66 (53.2)	1.309 [0.7132–2.402]	0.3844
CC		20 (19.1)	26 (21)	1.341 [0.6274–2.865]	0.4487
Alleles					
T		118 (56.2)	130 (52.4)	1 ^1^	
C		92 (43.8)	118 (47.6)	1.164 [0.8046–1.685]	0.4196
**rs1903003**					
Genotypes					
TT		17 (16.2)	37 (29.8)	1 ^1^	
CT		51 (48.6)	65 (52.5)	0.5856 [0.2962–1.158]	0.1219
CC		37 (35.2)	22 (17.7)	0.2732 [0.1252–0.5960]	0.0009 *
Alleles					
T		85 (40.5)	139 (56)	1 ^1^	
C		125 (59.5)	109 (44)	0.5332 [0.3673–0.7740]	0.0009 *
**rs2869967**					
Genotypes					
TT		27 (25.8)	34 (27.4)	1 ^1^	
CT		52 (49.5)	65 (52.4)	0.9926 [0.5323–1.851]	0.9815
CC		26 (24.7)	25 (20.2)	0.7636 [0.3620–1.610]	0.4782
Alleles					
T		106 (50.5)	133 (53.6)	1 ^1^	
C		104 (49.5)	115 (46.4)	0.8813 [0.6099–1.274]	0.5009

^1^ Reference category; OR (95% CI), odds ratio (95% confidence interval). * Result statistically significant.

**Table 3 genes-14-00914-t003:** Anthropometric and blood pressure analysis [expressed as Mean ± SD] in patients with co-existing genotypes of *FAM13A* (rs1903003, rs7671167, rs2869967) with *PPARγ-2 gene* genotypes (rs3856806, C1431T) prevalence.

Anthropometric and Blood Pressure Parameters [Mean ± SD]					
*FAM13A* rs1903003 (CT)					
*PPAR-γ2*	BMI [kg/m^2^]	Body Mass [kg]	FBM [% Body Mass]	LBM [% Body Mass]	
C1431T (*n* = 32)	29.7 ± 8.3	77.8 ± 21.2	43.5 ± 7.8	56.5 ± 7.8	
C1431C (*n* = 81)	29.4 ± 6.0	75.8 ± 16.3	43.2 ± 6.4	56.8 ± 6.4	
T1431T (*n* = 3)	31.8 ± 1.5	85.6 ± 5.9	46.8 ± 2.1	53.2 ± 2.1	
	**BMR [kcal]**	**SBP [mmHg]**	**DBP [mmHg]**		
C1431T (*n* = 32)	1371 ± 146	87 ± 15	142 ± 27	0.83 ± 0.09	
C1431C (*n* = 81)	1363 ± 136	90 ± 14	142 ± 22	0.84 ± 0.08	
T1431T (*n* = 3)	1436 ± 108	114 ± 10	181 ± 12	0.90 ± 0.02	
***FAM13A* rs1903003 (TT)**					
	**BMI [kg/m^2^]**	**Body mass [kg]**	**FBM [% body mass]**	**LBM**	
C1431T (*n* = 17)	26.4 ± 6.09	69.3 ± 11.1	40.9 ± 6.9	59.1 ± 6.7	
C1431C (*n* = 57)	29.3 ± 5.61	76.9 ± 15.1	43.6 ± 5.8	56.5 ± 5.8	
T1431T (*n* = 0)	*p* = 0.0173				
	**BMR [kcal]**	**SBP [mmHg]**	**DBP [mmHg]**	**WHR**	
C1431T (*n* = 17)	1317 ± 102	87 ± 11	147 ± 19	0.7 ± 0.06	
C1431C (*n* = 57)	1367 ± 153	86 ± 14	138 ± 23	0.8 ± 0.07	
T1431T (*n* = 0)				*p* = 0.0068	
***FAM13A* rs1903003 (CC)**					
	**BMI [kg/m^2^]**	**Body mass [kg]**	**FBM [% body mass]**	**LBM [% body mass]**	
C1431T (*n* = 17)	27.3 ± 6.4	70.9 ± 16.9	40.7 ± 8.0	59.3 ± 8.0	
C1431C (*n* = 57)	29.5 ± 5.3	75.7 ± 12.5	43.6 ± 6.3	56.4 ± 6.3	
T1431T (*n* = 0)	27.5 ± 3.6	73.2 ± 16.8	41.8 ± 3.0	58.3 ± 3.0	
	**BMR [kcal]**	**SBP [mmHg]**	**WHR**		
C1431T (*n* = 17)	1334 ± 104	149 ± 16	90 ± 9	0.84 ± 0.08	
C1431C (*n* = 57)	1360 ± 96	135 ± 22	86 ± 13	0.83 ± 0.08	
T1431T (*n* = 0)	1365 ± 166	129 ± 30	86 ± 21	0.91 ± 0.09	
		*p* = 0.0279			
***FAM13A* rs2869967 (CC)**			***FAM13A* rs1903003 (CC)**		
	**BMI [kg/m^2^]**	**WHR**		**SBP [mmHg]**	**DBP [mmHg]**
C1431T (*n* = 16)	26.9 ± 6.37	0.80 ± 0.07	Trp64Trp (61)	142.1 ± 21.3	89.6 ± 11.9
C1431C (*n* = 35)	29.3 ± 5.73	0.86 ± 0.06	Trp64Arg (12)	125 ± 15.6	79 ± 6.53
T1431T (*n* = 0)	-	-	Arg64Arg (0)	-	-
	*p* = 0.0494	*p* = 0.0076		*p* = 0.0275	*p* = 0.0480

*n*—number of patients; X—mean; SD—standard deviation; BMI—body mass index; WHR—waist to hip ratio; LBM—lean body mass [% of total body mass]; FBM—fat body mass [% of total body mass]; BMR—basal metabolic rate; SBP—systolic blood pressure; DBP—diastolic blood pressure.

**Table 4 genes-14-00914-t004:** Characteristics of fat distribution (WHR) in lean and obese women.

Analysis of Fat Distribution (WHR) According to BMI [kg/m^2^] Value in the Co-Existence of Selected Polymorphisms				
		*FAM13A* Polymorphisms		
		rs1903003 (CT)	rs1903003 (TT)	rs1903003 (CC)
	*PPAR-γ2* polymorphism	WHR [Mean ± SD]		
BMI < 25 kg/m^2^	C1431T (*n* = 15)	0.77 ± 0.08	0.76 ± 0.03	0.82 ± 0.06
	C1431C (*n* = 36)	0.8 ± 0.08	0.81 ± 0.08	0.81 ± 0.07
	*p*-value	ns	ns	ns
BMI > 30 kg/m^2^	C1431T (*n* = 17)	0.88 ± 0.06	0.87 ± 0.03	0.87 ± 0.09
	C1431C (*n* = 45)	0.87 ± 0.06	0.87 ± 0.04	0.84 ± 0.08
	*p*-value	ns	ns	ns
		rs7671167 (CT)	rs7671167 (TT)	rs7671167 (CC)
BMI < 25 kg/m^2^	C1431T (*n* = 12)	0.77 ± 0.08	0.76 ± 0.03	0.82 ± 0.06
	C1431C (*n* = 25)	0.8 ± 0.08	0.82 ± 0.08	0.805 ± 0.07
	*p*-value	ns	ns	ns
BMI > 30 kg/m^2^	C1431T (*n* = 5)	0.88 ± 0.06	0.87 ± 0.03	0.87 ± 0.09
	C1431C (*n* = 32)	0.87 ± 0.05	0.87 ± 0.04	0.84 ± 0.08
	*p*-value	ns	ns	ns
		rs2869967 (CT)	rs2869967 (TT)	rs2869967 (CC)
BMI < 25 kg/m^2^	C1431T (*n* = 10)	0.77 ± 0.07	0.81 ± 0.06	0.76 ± 0.03
	C1431C (*n* = 16)	0.80 ± 0.08	0.78 ± 0.06	0.84 ± 0.07
	*p* value	ns	ns	*p* < 0.0481
BMI > 30 kg/m^2^	C1431T (*n* = 6)	0.88 ± 0.07	0.88 ± 0.08	0.87 ± 0.03
	C1431C (*n* = 19)	0.87 ± 0.05	0.86 ± 0.08	0.87 ± 0.05
	*p*-value	ns	ns	ns

N—number of patients; X—mean; SD—standard deviation; ns—statistically not significant.

## Data Availability

The data presented in this study are available on request and approval from the corresponding author.

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
