# Peer review of "The Influence of FAM13A and PPAR-γ2 Gene Polymorphisms on the Metabolic State of Postmenopausal Women"

_genes, 2023, doi:10.3390/genes14040914_

Round 1

Reviewer 1 Report

The Manuscript is dealing with investigation of gene variants in metabolic state  of women. Major: It requires extensive re-writing for better understanding of the text. 

Minor: 1. what is understanding of hypertension according to the authors: Hypertension is a major cardiovascular disease risk factor (line 38), cardiovascular diseases (including hypertension) (line 54), exclusion criteria  cardiovascular diseases (line 102), Is it means that patients with hypertension were not included into the study? 

2. More detail presentation of Statistical analysis.: distribution of the data, multiple comparison test for gene variants statistics.

3. What is meaning of 182-184 lines.

4. Tables require edition to make easier understanding of the data. What is Me?   

5. Discussion is superficially addressed. No biological meaning of the results is presented.

Author Response

Dear Reviewer,

Thank you very much for your effort and revision of our manuscript. All comments have been considered and adequately corrected in the text and tables (red color) – please see attached document. 

Kind regards

Bogna Grygiel-Górniak 

Reviewer 2 Report

Grygiel-Górniak et al. analyzed genetic variants, fat distribution, and hypertension risk in lean and obese metabolically healthy postmenopausal women. Interestingly, the authors showed that individual SNP does not determine the fat amount/distribution, but rather identified a constellation of genetic variants in the PPAR-γ2 gene with several FAM13A gene polymorphisms related to higher BMI values and visceral fat distribution. Also, the co-association of rs1903003 (CC) in FAM13A and Trp64Arg in 3β-AR was related to hypertension in obese premenopausal women. The results of this work are interesting for scientists studying the molecular basis and the roles of FAM13A, PPAR, and 3β-AR in visceral obesity, as well as for medical doctors because of the important role of the genetic background of obesity and its complications.

The study is well-designed, and the applied methods are appropriate and informative. The results are clearly presented with the appropriate tables.

I would like to give some minor suggestions for the improvement of your manuscript:

- Delete the first paragraph in the Results section (Lines 182-184).

- For better understanding please add the WHR in the Table 4 Caption. It should be Analysis of fat distribution (WHR) according to BMI value in co-existence of selected polymorphisms

-There are several spelling errors in Table 3 and 4 legends (Lines 222-225; 228)

-Check the format and spelling errors in the tables. Delete extra spaces (Table 4), the mark for the square meter (kg/m2) should be in superscript (Table 3)...

-Use abbreviations for full names at the first appearance in the text: a genome-wide association studies (GWAS), PW patients, etc., or just full names.

- I just wonder why the authors put inflammation as the focus of their discussion. Low-grade systemic inflammation/meta inflammation is a consequence of adipocyte dysfunction, and this should be elaborated on in more detail.

- At the end of the discussion, it should be interesting to underline the perspectives of investigations of coexistences of several genetic variants in PPAR-γ2, FAM13A, and 3β-AR genes with respect to metabolic obesity, in normal-weight, metabolically obese subjects.

Author Response

(The authors gave the same response as above.)

Reviewer 3 Report

There are the following comments/suggestions on the reviewed article: 1) genes should be indicated in italics; 2) were their functional significance (epigenetic modifications, connection with expression, etc.) taken into account when selecting polymorphisms for research?; 3) were any covariates used in the analysis of associations?; 4) were corrections for multiple comparisons introduced in the analysis of associations?5) it is necessary to provide data on the presence in women of the studied groups of somatic pathology, including diseases of the female reproductive organs; 6) for the purpose of biomedical examination of the identified associations, it is necessary to analyze the regulatory potential of "significant" polymorphisms (link with transcriptional factors, modified histones, effect on gene expression, etc.), including in adipose tissue. This study can be carried out in silico.

Author Response

(The authors gave the same response as above.)

Round 2

Reviewer 1 Report

1. Materials and Methods: "Subjects receiving concomitant medication, including lipid lowering and hypoglycemic medications, were excluded from this analysis" . What about antihypertensive treatment? 

2.    PPAR-γ2 (Pro12Ala, rs1801282, and C1431T, rs3856806),  3β-AR (Trp64Arg, rs4994), FAM13A(rs1903003, rs7671167, rs2869967) were studied. Commonly in the text is presented only reference SNP number. It makes very complicated following the text. It means that you must come back to the methods.

3.  Table 3 and 4 requires edition. Title of Table 3 requires edition: to indicate genes typical for metabolic syndrome instead of gene variants only. 

4.Better structuring of the Discussion is recommended. The second paragraph is redundant as data on inflammatory state in obesity were not presented in the  manuscript. 

5. It is very difficult to follow discussion as in the text is commonly presented only reference SNP number. 

Author Response

Dear Editor,

Dear Reviewers,

Thank you very much for your effort and revision of our manuscript. All comments have been considered and adequately corrected in the text and tables (red color). Kind regardsBogna Grygiel-Górniak

Comments and Suggestions for Authors

  1. Materials and Methods: "Subjects receiving concomitant medication, including lipid lowering and hypoglycemic medications, were excluded from this analysis" . What about antihypertensive treatment? 

Thank you for your suggestion. Finding obese postmenopausal women without cardiovascular diseases was very difficult. Thus, we excluded from the study every woman with cardiovascular disease (e.g., myocardial infarction, cardiac insufficiency, arterial fibrillation or another arrhythmia, dyslipidemia, etc.) with the exception of hypertension. Since the patients with hypertension participated in this study, every patient with hypertension has been treated. Therefore, hypotensive drugs have not been excluded.

  1.   PPAR-γ2 (Pro12Ala, rs1801282, and C1431T, rs3856806),  3β-AR (Trp64Arg, rs4994), FAM13A(rs1903003, rs7671167, rs2869967) were studied. Commonly in the text is presented only reference SNP number. It makes very complicated following the text. It means that you must come back to the methods.

SNP reference numbers have been supplemented with gene names to improve the clarity of the text.

  1. Table 3 and 4 requires edition. Title of Table 3 requires edition: to indicate genes typical for metabolic syndrome instead of gene variants only. 

Tables 3 and 4 have been modified according to your suggestions

Table 3. Anthropometric and blood pressure analysis [expressed as Mean ± SD] in patients with co-existed genotypes of FAM13A (rs1903003, rs7671167, rs2869967) with PPARγ-2 gene genotypes (rs3856806, C1431T) prevalence.

Anthropometric and blood pressure parameters [Mean ± SD]

FAM13A rs1903003 (CT)

PPARγ2

BMI [kg/m2]

Body mass [kg]

FBM [%body mass]

LBM  [%body mass]

C1431T (n=32)

29.7  ± 8.3

77.8 ± 21.2

43.5 ± 7.8

56.5 ± 7.8

C1431C (n=81)

29.4 ± 6.0

75.8 ± 16.3

43.2 ± 6.4

56.8± 6.4

T1431T (n=3)

31.8 ± 1.5

85.6 ± 5.9

46.8 ± 2.1

53.2 ± 2.1

BMR [kcal]

SBP [mmHg]

DBP [mmHg]

C1431T (n=32)

1371 ± 146

87 ± 15

142 ± 27

0.83 ± 0.09

C1431C (n=81)

1363 ± 136

90 ± 14

142 ± 22

0.84 ± 0.08

T1431T (n=3)

1436 ± 108

114 ± 10

181 ± 12

0.90 ± 0.02

FAM13A rs1903003 (TT)

BMI [kg/m2]

Body mass [kg]

FBM [%body mass]

LBM

C1431T (n=17)

26.4 ± 6.09

69.3± 11.1

40.9± 6.9

59.1 ± 6.7

C1431C (n=57)

29.3± 5.61

76.9± 15.1

43.6± 5.8

56.5 ± 5.8

T1431T (n=0)

p=0.0173

BMR [kcal]

SBP [mmHg]

DBP [mmHg]

WHR

C1431T (n=17)

1317 ± 102

87 ± 11

147 ± 19

0.7± 0.06

C1431C (n=57)

1367 ± 153

86 ± 14

138 ± 23

0.8± 0.07

T1431T (n=0)

p=0.0068

FAM13A rs1903003 (CC)

BMI [kg/m2]

Body mass [kg]

FBM [%body mass]

LBM [%body mass]

C1431T (n=17)

27.3 ± 6.4

70.9  ± 16.9

40.7  ± 8.0

59.3  ± 8.0

C1431C (n=57)

29.5 ± 5.3

75.7  ± 12.5

43.6  ± 6.3

56.4  ± 6.3

T1431T (n=0)

27.5 ± 3.6

73.2  ± 16.8

41.8  ± 3.0

58.3  ± 3.0

BMR [kcal]

SBP [mmHg]

WHR

C1431T (n=17)

1334  ± 104

149  ± 16

90  ± 9

0.84 ± 0.08

C1431C (n=57)

1360  ± 96

135  ± 22

86  ± 13

0.83 ± 0.08

T1431T (n=0)

1365  ± 166

129  ± 30

86  ± 21

0.91 ± 0.09

p=0.0279

FAM13A rs2869967 (CC)

FAM13A rs1903003 (CC)

BMI [kg/m2]

WHR

SBP [mmHg]

DBP [mmHg]

C1431T (n=16)

26.9 ± 6.37

0.80 ± 0.07

Trp64Trp (61)

142.1 ± 21.3

89.6 ± 11.9

C1431C (n=35)

29.3 ± 5.73

0.86 ± 0.06

Trp64Arg (12)

125 ± 15.6

79 ± 6.53

T1431T (n=0)

-

-

Arg64Arg (0)

-

-

p=0.0494

p=0.0076

p=0.0275

p=0.0480

N-number of patients; X – mean; SD – standard deviation; BMI – body mass index; WHR – waist to hip ratio; LBM – lean body mass [% of total body mass]; FBM – fat body mass [% of total body mass]; BMR – basal metabolic rate; SBP – systolic blood pressure; DBP – diastolic blood pressure.

Table 4. Characteristics of fat distribution (WHR) in lean and obese women.

Analysis of fat distribution (WHR) according to BMI [kg/m2] value in the co-existence of selected polymorphisms

FAM13A polymorphisms

rs1903003 (CT)

rs1903003

(TT)

rs1903003

(CC)

PPARγ2

polymorphism

WHR [Mean ± SD]

BMI<25 kg/m2

C1431T (n=15)

0.77 ± 0.08

0.76 ± 0.03

0.82 ± 0.06

C1431C (n=36)

0.8 ± 0.08

0.81 ± 0.08

0.81 ± 0.07

p-value

ns

ns

ns

BMI>30 kg/m2

C1431T (n=17)

0.88 ± 0.06

0.87 ± 0.03

0.87 ± 0.09

C1431C (n=45)

0.87 ± 0.06

0.87 ± 0.04

0.84 ± 0.08

p-value

ns

ns

ns

rs7671167

(CT)

rs7671167

(TT)

rs7671167

(CC)

BMI<25 kg/m2

C1431T (n=12)

0.77 ± 0.08

0.76 ± 0.03

0.82 ± 0.06

C1431C (n=25)

0.8  ± 0.08

0.82 ± 0.08

0.805 ± 0.07

p-value

ns

ns

ns

BMI > 30 kg/m2

C1431T (n=5)

0.88 ± 0.06

0.87 ± 0.03

0.87 ± 0.09

C1431C (n=32)

0.87 ± 0.05

0.87 ± 0.04

0.84 ± 0.08

p-value

ns

ns

ns

rs2869967

(CT)

rs2869967

(TT)

rs2869967

(CC)

BMI<25 kg/m2

C1431T (n=10)

0.77 ± 0.07

0.81 ± 0.06

0.76 ± 0.03

C1431C (n=16)

0.80 ± 0.08

0.78 ± 0.06

0.84 ± 0.07

P value

ns

ns

p<0.0481

BMI>30 kg/m2

C1431T (n=6)

0.88 ± 0.07

0.88 ± 0.08

0.87 ± 0.03

C1431C (n=19)

0.87 ± 0.05

0.86 ± 0.08

0.87 ± 0.05

p-value

ns

ns

ns

N-number of patients; X – mean. SD – standard deviation; ns – statistically not significant

4.Better structuring of the discussion is recommended. The second paragraph is redundant as data on inflammatory state in obesity were not presented in the  manuscript. 

Thank you for your remark.

The paragraph about low-grade inflammation has been added because one of the reviewers wrote:

“I just wonder why the authors put inflammation as the focus of their discussion. Low-grade systemic inflammation/meta inflammation is a consequence of adipocyte dysfunction, and this should be elaborated on in more detail”. Thus, we answered this request and added information regarding low-grade inflammation. However, we shortened this paragraph a bit.

In obesity, enlarged adipocytes press the surrounding vessels and adipose tissue cells and cause local hypoxia (39). As a result, increased fibrotic processes and infiltration of macrophages are observed. Macrophages scavenge necrotic adipocytes and trigger a local inflammatory response. They increase the synthesis of pro-inflammatory cytokines such as TNF-a and IL-6 (39). The inflammatory state in obesity is also related to the transformation of M2 macrophages (anti-inflammatory) to M1 macrophages (pro-inflammatory). Such conversion correlates with insulin resistance (40). Low-grade inflammation in adipose tissue is also stimulated by B-cells, which infiltrate adipose tissue and synthesize various cytokines and antibodies (IgG) (40) (41).

  1. It is very difficult to follow discussion as in the text is commonly presented only reference SNP number. 

Thank you for your remark. The SNP reference number is the most commonly used way of naming polymorphisms. With a large number of SNPs under study, this can make it difficult to keep the discussion transparent. We have added the gene name to each “rs” to make the text more readable.
